

# Detailed Analysis of the Blade Root Flow of a Horizontal Axis Wind Turbine

I. Herráez[1], B. Akay[2], G. J. W. van Bussel[2], J. Peinke[1,3], and B. Stoevesandt[3]

[1]Institute of Physics, University of Oldenburg, 26111, Oldenburg, Germany
[2]Faculty of Aerospace Engineering, Delft University of Technology, 2629HS Delft,
the Netherlands
[3]Fraunhofer IWES, Ammerländer Heerstr. 136, 26129, Oldenburg, Germany

Received: 12 December 2015 – Accepted: 16 December 2015 – Published: 19 January 2016

Correspondence to: I. Herráez (ivan.herraez@forwind.de)

Published by Copernicus Publications on behalf of the European Academy of Wind Energy
e.V.



**Abstract**

The root flow of wind turbine blades is subjected to complex physical mechanisms that influence significantly the rotor aerodynamic performance. Spanwise flows, the Himmelskamp effect and the formation of the root vortex are examples of interre-
5 lated aerodynamic phenomena observed in the blade root region. In this study we address those phenomena by means of Particle Image Velocimetry (PIV) measure-ments and Reynolds Averaged Navier–Stokes (RANS) simulations. The numerical re-sults obtained in this study are in very good agreement with the experiments and un-veil the details of the intricate root flow. The Himmelskamp effect is shown to delay the
10 stall onset and enhance the lift force coefficient $C_l$ even at a moderate angle of attack (AoA $\approx 13°$). The results also show that the vortex emanating from the spanwise posi-tion of maximum chord length rotates in the opposite direction of the root vortex, what affects the wake evolution.

## 1 Introduction

The aerodynamic design of wind turbine blades is subjected to important levels of un-certainty. As a matter of fact, not only transient operational states can pose a challenge to the wind turbine designer, but also seemingly simple cases involving steady opera-tion under axisymmetric, uniform inflow conditions (Leishman, 2002; Schepers, 2012). This is especially true for the tip and root regions of the blades, where the flow is three-
dimensional and strongly influenced by the trailing vortices (Micallef, 2012).

**Spanwise flows and Himmelskamp effect**

At the root of the blade, the angle of attack (AoA) is usually considerably higher than at the tip. This increases the complexity of the flow, since it often leads to flow sep-aration, what in this part of the blade generally gives rise to the Himmelskamp effect
(Himmelskamp, 1947). This effect delays the stall onset and enhances the lift force as

Discussion Paper | Discussion Paper | Discussion Paper | Discussion Paper |

**WESD**

doi:10.5194/wes-2015-1

**Analysis of the Blade Root Flow**

I. Herráez et al.

**WESD**

doi:10.5194/wes-2015-1

**Analysis of the Blade Root Flow**

I. Herráez et al.

compared to non-rotating blades operating at the same AoA. The Himmelskamp effect, also known as *stall delay* or *rotational augmentation*, has been studied by many authors both experimentally (Schreck and Robinson, 2002; Sicot et al., 2008; Ronsten, 1992) and numerically (Guntur and Sørensen, 2015; Herráez et al., 2014; Schreck

et al., 2007), although it still remains far from being well understood and characterized. It mainly affects the blade root region and is known to be closely related to the existence of spanwise flows in the boundary layer. Snel et al. (1993) were the first to propose a correction model to be applied to 2-D airfoil characteristics in order to account for this effect in Blade Element Momentum (BEM) and other engineering tools that rely in 2-D

airfoil data. More correction models have been developed since then (Chaviaropoulos and Hansen, 2000; Bak et al., 2006; Raj, 2000; Corrigan and Schillings, 1994, etc.). However, Breton et al. (2008) and Guntur et al. (2011) proved that their accuracy is still a critical issue. Currently, a major impediment in the development of accurate correction models is the incomplete understanding of the physical mechanisms. It is worth

highlighting that, up to now, the study of the Himmelskamp effect has been mostly focused on post-stall conditions. Consequently, very little is known about its onset at moderate angles of attack.

**The root vortex**

One fundamental feature of the root (and tip) flow is the formation of trailing vorticity
that rolls up into a discrete vortex. Several authors have attempted to capture experimentally the root vortex in the near wake of a wind turbine. However, as Vermeer et al. (2003) highlighted, this can be extremely difficult to achieve due to the fact that the near wake usually does not present a distinctive, well defined root vortex (opposite to the tip vortex). Many wind tunnel experiments with model wind turbines confirmed this. For
instance, Massouh and Dobrev (2007) and Haans et al. (2008) also came to that conclusion after studying a wind turbine rotor wake with Particle Image Velocimetry (PIV) and hot film wake measurements, respectively. Furthermore, Ebert and Wood (2001) and Sherry et al. (2013) observed by means of PIV (among other measurement tech-

niques) that the root vortex diffuses very rapidly. The PIV measurements performed by Akay et al. (2012) on two different rotors demonstrated that the evolution and strength of the root vortex highly depends on the blade root geometry and the spanwise distribution of circulation. In a subsequent work also based on PIV measurements, Akay

et al. (2014) suggested that the flow in the root region is driven by the bound vorticity.

The study of the root (and tip) vortices can also be addressed by means of numerical simulations. For this purpose, Large Eddy Simulations (LES) are commonly combined with actuator line models (Ivanell et al., 2007; Troldborg et al., 2007; Nilsson et al., 2015). This technique is very useful for analysing the evolution of the trailing vortices

in the wake.

However, it implies a very strong simplification of the blade geometry, what makes it unsuitable for studying the origin of the root and tip vortices. This is well exemplified in van Kuik et al. (2014), where it is concluded that the fact that actuator line models disregard the chordwise bound circulation at the blade tip prevents them from comput-

ing correctly the tip vortex trajectory in the vicinity of the blade. The same article also shows that full blade Reynolds Averaged Navier–Stokes Simulations (RANS) as well as panel code computations allow a much more realistic study of the tip vortex formation mechanism. Indeed, the use of a panel code allowed Micallef et al. (2012) to study the origin of the tip vortex on a wind tunnel model rotor, unveiling the complex distribution

of bound vorticity at the blade tip. However, to the best of our knowledge, the origin of the root vortex has still not been addressed in detail up to now.

**Scope and outline**

This article aims at gaining insight both experimentally and numerically into the root flow of a horizontal axis wind turbine operating at design conditions. The focus is put

into two important and interrelated aspects of the root flow that, as above explained, are insufficiently understood so far:

**WESD**

doi:10.5194/wes-2015-1

**Analysis of the Blade Root Flow**

I. Herráez et al.

1. Spanwise flows and onset of the Himmelskamp effect at moderate angles of attack (design operating conditions)

2. The origin of the root vortex

Section 2.1 and 2.2 describe the experimental and numerical set-up, respectively. The main characteristics of the flow over the root region are presented in Sect. 3.1. Furthermore, in this section the simulations are validated against experimental results. The presence of spanwise flows is further discussed in Sect. 3.2. Section 3.3 addresses the onset of the Himmelskamp effect. The origin of the root vortex is analysed in Sect. 3.4. Finally, the main conclusions of this work are summarized in Sect. 4.

## 2   Methods

### 2.1   Experimental setup

The scope of the experimental campaign is to measure the three components of the flow over the root region of a wind turbine blade. This is achieved by means of stereoscopic Particle Image Velocimetry (PIV).

The measurements are carried out in the Open Jet Facility of the Faculty of Aerospace Engineering at the Delft University of Technology. This wind tunnel has an octagonal open jet with an equivalent diameter of 3 m. The studied wind turbine consists of a two-bladed rotor with a diameter of 2 m. The chord and twist distributions are shown in Fig. 1. Table 1 shows the used airfoil types.

The measurement campaign includes both a spanwise and a chordwise configuration of the PIV windows. The spanwise measurements are carried out at different azimuth angles for capturing the evolution of the near wake. In the chordwise configuration the PIV windows are orthogonal to the blade axis around the blade chord. This configuration, which included 40 different radial positions, offers the best insight into the flow around the blade surface.

**WESD**

doi:10.5194/wes-2015-1

**Analysis of the Blade Root Flow**

I. Herráez et al.

Discussion Paper | Discussion Paper | Discussion Paper | Discussion Paper

The measurements are phase-locked and phase-averaged with the azimuthal position of the rotor blade rotation. This allows to reconstruct the flow over each blade section after measuring the pressure and suction sides separately.

The rotor operated at rated conditions with a freestream wind speed $U_\infty = 6\,\mathrm{m\,s^{-1}}$ and a rotational speed $\omega = 400\,\mathrm{rpm}$ (tip speed ratio $\lambda = 7$). The turbulence intensity is $TI = 0.28\,\%$ and there is no yaw misalignment. The Reynolds number at the radial position of maximum chord reached $Re \approx 1.5 \times 10^5$.

Further details about the experimental set-up can be found in Akay et al. (2014).

## 2.2 Numerical method and computational mesh

The simulations presented in this work are based on the Reynolds-Averaged Navier–Stokes (RANS) method and they are performed with the open source code OpenFOAM (www.openfoam.com). The computational model solves the incompressible Navier–Stokes equations using a finite volume approach for the spatial discretization. The convective terms are discretized with a second order linear-upwind scheme. For the viscous terms a second-order central differences linear scheme is employed. The use of a non-inertial reference frame and the addition of the Coriolis and centrifugal forces to the momentum equations allows to account for the rotation of the system. The SIMPLE algorithm is employed for enforcing the pressure-velocity coupling. The turbulence in the boundary layer is modelled by means of the $k - \omega$ Shear-Stress Transport (SST) model proposed by Menter (1993).

The grid is generated with the software Pointwise (www.pointwise.com). It exploits the symmetry of the rotor by modelling only one half of it and using periodic boundary conditions. The computational domain is represented in Fig. 3 and it consists of two independent block-structured grids connected by means of a so called arbitrary mesh interface. The outer grid is a semi-sphere with the radius $22\,R$, where $R$ is the blade radius. The inner grid, which contains the blade, is a cylinder with the radius $1.1\,R$ and the height $1.1\,R$. The motivation for using two structured grids connected by an interface is to independently control the mesh resolution in the proximity of the blade

**WESD**

doi:10.5194/wes-2015-1

**Analysis of the Blade Root Flow**

I. Herráez et al.

**WESD**

doi:10.5194/wes-2015-1

**Analysis of the Blade Root Flow**

I. Herráez et al.

and in the far field. The total number of cells is $9.8 \times 10^6$. The blade surface mesh (see Fig. 3) contains 130 cells along the chord, while 210 cells are used in the spanwise direction. In order to properly resolve the boundary layer, the height of the first cell in the normal direction to the blade surface is set to $5 \times 10^{-6}$ m, what ensures that $Y+$ is smaller than one along the whole blade. The semi-spherical outer boundary employs a boundary condition that changes its behaviour depending on the direction of the flow: in regions where the flow goes in, it works like a Dirichlet boundary condition assuming a predefined value of the velocity field. In regions where the flow goes out, it enforces a zero gradient condition (Neumann condition). The symmetry plane made use of periodic boundary conditions. Non-slip boundary conditions were applied to the blade surface.

## 3   Results and discussion

### 3.1   Main characteristics of the flow field over the blades

The detailed PIV and numerical results provide a good insight into the main flow characteristics over the blade root. It is important to note that the PIV data are partly affected by (sickle-shaped) reflection artefacts in front of the leading edge. Those artefacts are easily recognizable in Figs. 4, 5 and 6, and they will be just neglected in the interpretation of the results.

Figure 4 shows the azimuthal velocity component in an inertial reference frame for the radial stations $r = 0.26\,R$, $r = 0.35\,R$ (the position of maximum chord length) and $r = 0.45\,R$. The results are normalized with the free-stream wind speed $U_\infty$. The agreement between experimental and numerical results is excellent for all the studied radial positions. At $r/R = 0.35$ and $r/R = 0.45$, the azimuthal velocities are positive over the whole suction side except in the trailing edge region. However, at $r = 0.26\,R$ the suction side presents negative velocities from the mid-chord until the trailing edge. This does not necessarily imply flow separation and recirculation, though, since the relative

Discussion Paper | Discussion Paper | Discussion Paper | Discussion Paper |

velocity might still remain positive along the whole suction side. Indeed, at that radial position the local circumferential velocity caused by the blade rotation is $1.82/U_\infty$, what implies that the flow remains attached up to $U_\theta = -1.82/U_\infty$. In Sect. 3.3 the lack of separation is demonstrated by means of the wall shear stress.

The axial flow component is shown in Fig. 5. The axial velocity over the second half of the suction side becomes smaller with increasing radial position. This is in fact just a geometrical effect that occurs as a consequence of the twist of the rotor blade. At $r = 0.45\,R$ the orientation of the second half of the suction side is slightly upstream, so the axial velocity becomes negative if the flow is attached. The effect is smaller for lower radial positions because of the larger twist angle, which neutralizes the mentioned geometrical effect. The numerical results are consistent with the experiments, although at $r = 0.45\,R$ the axial velocity over the suction side is slightly overpredicted.

Figure 6 displays the radial velocity component for the three considered spanwise positions. At $r = 0.26\,R$, this velocity is very strong on the suction side. However, at $r = 0.35\,R$ and $r = 0.45\,R$ it becomes much weaker. Hence, the presence of spanwise flows seems to be limited to the innermost region of the blade. The agreement between experiments and simulations is again very good for all three stations.

The velocity field 10 mm off the suction side has been extracted from both the numerical results and the available PIV data (including 40 different radial positions between $r = 0.17\,R$ and $r = 0.65\,R$) in order to study the flow in the proximity of the blade surface in more detail. Figure 7 shows that the azimuthal component is always positive outboard of the position of maximum chord length ($r = 0.35\,R$). Below that position, a significant region of the blade presents negative azimuthal velocities close to the trailing edge. At $r \approx 0.3\,R$, this effect is stronger in the PIV measurements than in the numerical results. Apart from this, the numerical results are in very good agreement with the experimental results.

The axial flow velocity component is displayed in Fig. 8. Outboard of the radial position of maximum chord length, the axial velocity becomes negative from the mid-chord towards the trailing edge. The same effect was already discussed in relation to Fig. 5.

**WESD**

doi:10.5194/wes-2015-1

**Analysis of the Blade Root Flow**

I. Herráez et al.

Interactive Discussion

Discussion Paper | Discussion Paper | Discussion Paper | Discussion Paper

The fact that the numerical results underpredict a bit this effect, which is caused by the relative position of the suction side to the rotor plane, might indicate a possible small deviation in the pitch angle. The agreement between PIV and CFD in the root region is very good, which is a clear indication that the wake and blade inductions are correctly
predicted with the current CFD model.

Figure 9 presents the distribution of radial velocity along the blade suction side. The experimental results show a substantial spanwise flow in the leading edge region from $r = 0.45\,R$ outwards. This is rather surprising, since the flow in that region is fully attached (as shown for instance in Fig. 7) and it is far away from the tip and root, where
spanwise flows are usually expected. The numerical results show much smaller radial velocities in that region. At present the authors do not have a solid explanation for this discrepancy. In the root region both PIV and CFD show evidence of strong spanwise flows in the proximity of the trailing edge, although the simulation tends to underpredict the spanwise flow in the region $0.3\,R < r < 0.35\,R$, as it also happened with the
azimuthal velocity (Fig. 7). As stated earlier, this might be caused by a slight deviation of the pitch angle. Other than this, the consistency between PIV and CFD is again very good, what gives confidence in the reliability of the numerical model in predicting the complex flows of the root region.

## 3.2 The source of the spanwise flows

Two different explanations have been proposed in the literature for explaining the origin of the spanwise flows:

1. *spanwise pressure gradients*: the dynamic pressure over the blade surface is inversely proportional to the radial position. Hence, the air is assumed to travel from the root towards outer positions as a consequence of spanwise pressure gradients (Schreck and Robinson, 2002; Schreck et al., 2010).

**WESD**

doi:10.5194/wes-2015-1

**Analysis of the Blade Root Flow**

I. Herráez et al.

Discussion Paper | Discussion Paper | Discussion Paper | Discussion Paper

2. *centrifugal force*: the centrifugal force that acts on the bottom of the boundary layer (i.e. the region where the flow is not detached from the surface) pushes the flow towards the tip (Lindenburg, 2003; Guntur and Sørensen, 2015).

In order to elucidate which is the actual source of the spanwise flows, the computed isobars of the blade suction side are compared with the limiting streamlines (obtained from the wall shear stress) in Fig. 10. As it can be seen, the surface pressure does not present significant spanwise gradients. It is worth remarking that the same observation was made in the analysis of the MEXICO wind tunnel experiment (Herráez et al., 2014). Therefore, we conclude that the centrifugal force is the main source of spanwise flows.

Figure 10 also shows how the Coriolis force progressively redirects the spanwise flow coming from the root towards the trailing edge, what makes the flow to follow a curved trajectory.

## 3.3 Onset of the Himmelskamp effect

Figure 11 compares the pressure coefficient $C_p$ distribution obtained from the blade at $r = 0.26\,R$ with the $C_p$ distribution extracted from a 2-D simulation at the same Reynolds number ($Re \approx 1 \times 10^5$) and same angle of attack (AoA $\approx 13°$, computed using the method proposed by Shen et al., 2009). The 2-D simulation is a RANS computation performed with the $k - \omega$ SST turbulence model. Also, the 2-D mesh is equivalent to the 3-D mesh except for the third dimension. Experimental results of the same 2-D airfoil with $Re = 1 \times 10^6$ are displayed as well. The 2-D experimental and numerical results present some disparity in the region of the suction peak, but apart from this they are very similar in spite of the difference of Reynolds number. However, the 3-D results exhibit some important differences. The slope of the adverse pressure gradient is substantially reduced, what leads to a delay of the separation point. The separation point can be approximately identified as the point where the adverse pressure gradient meets the region with zero pressure gradient (i.e. the region where the flow is separated). In the 2-D airfoils the separation point is located at $x/c \approx 0.4$. In the 3-D case,

**WESD**

doi:10.5194/wes-2015-1

**Analysis of the Blade Root Flow**

I. Herráez et al.

Discussion Paper | Discussion Paper | Discussion Paper | Discussion Paper | Discussion Paper

the adverse pressure gradient presents a kink at $x/c \approx 0.5$, but it stays negative for the whole chord length, what seems to indicate that the flow remains attached. However, Sicot et al. (2008) concluded that rotating blades can present separation even in regions of adverse pressure gradient. In order to verify if there is separation in the 3-D case, the skin friction coefficient $C_{fx}$ on the suction side is displayed in Fig. 12 for both the 2-D and 3-D simulations. In the 2-D case, $C_{fx}$ becomes positive at $x/c = 0.39$, indicating that the flow separates exactly at that point (in good agreement with the estimation from the $C_p$ distribution). In the 3-D case, $C_{fx}$ becomes zero at $x/c = 0.52$, but it recovers directly afterwards, remaining always negative. This confirms that the flow stays completely attached all along the chord. The point where $C_{fx}$ becomes zero is actually the place where the chordwise flow direction is deflected into a spanwise flow direction. The same happens for all other radial positions at the root. Therefore, the transition between the chordwise and spanwise flows in Fig. 10 can be considered as an isoline of $C_{fx} = 0$.

Another remarkable feature of the 3-D $C_p$ distribution from Fig. 11 is that both the pressure and suction sides present approximately the same slope shortly after the kink in the adverse pressure gradient ($x/c \approx 0.5$) until the trailing edge. This resembles the behaviour of the 2-D case in the region with zero pressure gradient. Finally, it is worth highlighting that the 3-D case presents a smaller suction peak than the 2-D case.

The resulting lift and drag coefficients ($C_l$ and $C_d$, respectively) for the 2-D and 3-D cases are presented in Table 2. $C_l$ is increased by approximately 9 % as a consequence of the Himmelskamp effect, whereas $C_d$ does not seem to be influenced at all. This is also in agreement with our observations from the MEXICO turbine, where the Himmelskamp effect had a very limited influence on the drag (Herráez et al., 2014).

## 3.4 The origin of the root vortex

The bound vorticity $\gamma$ can be computed as the difference in the velocity outside the boundary layer of the pressure and suction sides. $\gamma$ can then be decomposed into a

**WESD**

doi:10.5194/wes-2015-1

**Analysis of the Blade Root Flow**

I. Herráez et al.

Discussion Paper | Discussion Paper | Discussion Paper | Discussion Paper | Discussion Paper

radial $\gamma_{radial}$ and a chordwise $\gamma_{chordwise}$ component. Figure 13 shows both components side by side.

$\gamma_{radial}$ is concentrated around the 1/4 chord position, as might be expected. The radial circulation $\Gamma_{radial}$ can be computed from $\gamma_{radial}$ as its integral along the chord:

$$\Gamma_{radial} = \int_{le}^{te} \gamma_{radial} \cdot dx \tag{1}$$

where le and te are the leading and trailing edges, respectively. The use of the Kutta–Jukowski theorem allows then to compute the sectional lift:

$$L' = -\rho \cdot U_{rel} \cdot \Gamma_{radial} \tag{2}$$

where $\rho$ is the air density and $U_{rel}$ is the relative velocity in the plane perpendicular to the blade axis. Owing to the $\gamma_{radial}$ distribution from Fig. 13 and the strong link between $\gamma_{radial}$ and the lift, it can be concluded that the lift force is generated almost exclusively in the first half of the chord all along the blade. The decay of $\gamma_{radial}$, and hence the decay of the lift force, is much more sudden at the tip than at the root. As a consequence, the root losses take place much more gradually than the tip losses. This should be taken into account by the correction models used e.g. in the BEM and actuator line methods. $\gamma_{radial}$ is transformed into $\gamma_{chordwise}$ at the tip and root before becoming trailed free vorticity, what gives rise to the tip and root vortices. This is evidenced in Fig. 13b, where it can be seen that the tip and root regions present substantial $\gamma_{chordwise}$ in the proximity of the trailing edge. $\gamma_{chordwise}$ is distributed over a larger spanwise range at the root than at the tip, what is in agreement with the gradual root losses earlier described. van Kuik et al. (2014) obtained very similar results at the tip of a different rotor but the root was not studied. In the innermost region of the blade the sign of $\gamma_{chordwise}$ at the trailing edge is opposite to that of the tip (as one would expect from a horseshoe vortex model). However, in the region of maximum chord length, $\gamma_{chordwise}$ at the trailing edge presents the same sign as the tip vortex. The negative $\gamma_{chordwise}$ at the root implies

WESD

doi:10.5194/wes-2015-1

Analysis of the Blade Root Flow

I. Herráez et al.

an outward motion of the flow over the part of the suction side where the azimuthal velocity is slow (see Fig. 7). On the contrary, the positive $\gamma_{\text{chordwise}}$ in the region of maximum chord leads to an inward flow motion. Akay et al. (2014) studied the wake of the same wind turbine with PIV and indeed observed the presence of an outward flow for $r < 0.25\,R$ and the existence of an inward flow in the radial range $0.25\,R < r < 0.35\,R$. Furthermore, Medici and Alfredsson (2006) did similar observations in their experimental wake study of a different wind turbine. The present results not only confirm the mentioned experimental observations, but also explain the origin of this aerodynamic behaviour.

Figure 14 shows the bound vorticity vectors over the blade. From this figure it is evident how the direction of the bound vorticity $\gamma$ changes at the root. As it can be seen, at the most inboard part of the blade ($r < 0.35\,R$), $\gamma_{\text{chordwise}}$ dominates the flow over the second half of the chord, indicating that vorticity is trailed along that region.

The fact that $\gamma_{\text{chordwise}}$ is distributed over such a large area of the root might explain that the root vortex does not present a well defined, distinctive structure, as Vermeer et al. (2003), Massouh and Dobrev (2007) and Haans et al. (2008) reported in their experimental wake studies of different wind turbines. Furthermore, the existence of two adjacent root regions with counter-rotating $\gamma_{\text{chordwise}}$ might also explain the fast diffusion of the root vortex reported by Ebert and Wood (2001) and Sherry et al. (2013).

## 4   Conclusions

The use of Particle Image Velocimetry (PIV) measurements and Reynolds-Averaged Navier–Stokes (RANS) simulations enabled the analysis of the flow in the root region of a wind turbine blade operating at design conditions with axisymmetric inflow. The following conclusions are drawn:

–  The RANS method is capable of capturing accurately the main features of the root flow of wind turbine blades operating at design conditions.

Discussion Paper | Discussion Paper | Discussion Paper | Discussion Paper |

**WESD**

doi:10.5194/wes-2015-1

**Analysis of the Blade Root Flow**

I. Herráez et al.

- The spanwise flows in the root region are caused by the centrifugal force and not by radial pressure gradients, as some authors have suggested.

- Even at relatively moderate angles of attack (AoA $\approx 13°$), the interaction of the centrifugal and Coriolis forces can give rise to the Himmelskamp effect.

- The influence of the Himmelskamp effect on the sectional $C_p$ distribution is twofold: on one hand the suction peak is reduced, on the other hand the separation point is delayed (indeed in our case the separation is completely avoided). As a consequence of both counteracting effects, the influence of the Himmelskamp effect on the loads is weaker than in the $C_p$ distribution.

- The reduction of the aerodynamic performance is more gradual at the root than at the tip. Tip/root loss correction models (as used e.g. in BEM simulations) should account for this effect.

- The trailing vorticity in the spanwise position of maximum chord length presents the opposite sign than at the blade root. This contributes to the diffusion of the root vortex.

We recommend to consider these points for a better characterization of the root flow of wind turbine blades. This can help to reduce the uncertainty of the blade design process, what in turn would contribute to make the turbines more cost-effective.

*Author contributions.* I. Herráez performed the simulations. B. Akay carried out the measurements. G. J. W. van Bussel contributed to the physical interpretation of the experiments. B. Stoevesandt and J. Peinke supported the numerical work. All the authors participated in the analysis of the results. The manuscript was written by I. Herráez.

*Acknowledgements.* We thank the computer time provided by the Facility for Large-scale COmputations in Wind Energy Research (FLOW) at the University of Oldenburg.

Discussion Paper | Discussion Paper | Discussion Paper | Discussion Paper | Discussion Paper |

**WESD**

doi:10.5194/wes-2015-1

**Analysis of the Blade Root Flow**

I. Herráez et al.

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

## WESD

doi:10.5194/wes-2015-1

**Analysis of the Blade Root Flow**

I. Herráez et al.

Herráez, I., Stoevesandt, B., and Peinke, J.: Insight into Rotational Effects on a Wind Turbine Blade Using Navier–Stokes Computations, Energies, 7, 6798–6822, doi:10.3390/en7106798, 2014. 3, 10, 11

Himmelskamp, H.: Profile investigations on a rotating airscrew, Reports and translations, Völkenrode MAP, 1947. 2

Ivanell, S., Sørensen, J., Mikkelsen, R., and Henningson, D.: Numerical analysis of the tip and root vortex position in the wake of a wind turbine, J. Phys. Conf. Ser., 75, 012035, doi:10.1088/1742-6596/75/1/012035, 2007. 4

Leishman, J. G.: Challenges in modelling the unsteady aerodynamics of wind turbines, Wind Energy, 5, 85–132, doi:10.1002/we.62, 2002. 2

Lindenburg, C.: Investigation into rotor blade aerodynamics, Tech. Rep. ECN-C03-025, ECN, Petten, the Netherlands, 2003. 10

Massouh, F. and Dobrev, I.: Exploration of the vortex wake behind of wind turbine rotor, J. Phys. Conf. Ser., 75, 012036, doi:10.1088/1742-6596/75/1/012036, 2007. 3, 13

Medici, D. and Alfredsson, P. H.: Measurements on a wind turbine wake: 3D effects and bluff body vortex shedding, Wind Energy, 9, 219–236, doi:10.1002/we.156, 2006. 13

Menter, F.: Zonal two equation $k - \omega$ turbulence models for aerodynamic flows, AIAA Journal, 93, doi:10.2514/6.1993-2906, 1993. 6

Micallef, D.: 3D flows near a HAWT rotor: A dissection of blade and wake contributions, PhD thesis, Delft University of Technology, the Netherlands, 2012. 2

Micallef, D., Akay, B., Ferreira, C. S., Sant, T., and van Bussel, G.: The origins of a wind turbine tip vortex, in: The Science of Making Torque from Wind 2012, IOP Publishing, Oldenburg, Germany, 2012. 4

Nilsson, K., Shen, W. Z., Sørensen, J. N., Breton, S.-P., and Ivanell, S.: Validation of the actuator line method using near wake measurements of the MEXICO rotor, Wind Energy, 18, 1683–1683, doi:10.1002/we.1864, 2015. 4

Raj, N.: An improved semi-empirical model for 3-D post-stall effects in horizontal axis wind turbines, Master's thesis, University of Illinois, Urbana-Champaign, 2000. 3

Ronsten, G.: Static pressure measurements on a rotating and a non-rotating 2.375 m wind turbine blade. Comparison with 2D calculations, Journal of Wind Engineering and Industrial Aerodynamics, 39, 105–118, doi:10.1016/0167-6105(92)90537-K, 1992. 3

Schepers, J.: Engineering models in wind energy aerodynamics, PhD thesis, TU-Delft, 2012. 2

**WESD**

doi:10.5194/wes-2015-1

**Analysis of the Blade Root Flow**

I. Herráez et al.

Discussion Paper | Discussion Paper | Discussion Paper | Discussion Paper

Schreck, S. and Robinson, M.: Rotational augmentation of horizontal axis wind turbine blade aerodynamic response, Wind Energy, 5, 133–150, doi:10.1002/we.68, 2002. 3, 9

Schreck, S., Sant, T., and Micallef, D.: Rotational Augmentation Disparities in the MEXICO and UAE Phase VI Experiments, in: The Science of Making Torque from Wind 2010, Heraklion, Crete, Greece, 2010. 9

Schreck, S. J., Sørensen, N. N., and Robinson, M. C.: Aerodynamic structures and processes in rotationally augmented flow fields, Wind Energy, 10, 159–178, doi:10.1002/we.214, 2007. 3

Shen, W. Z., Hansen, M. O. L., and Sørensen, J. N.: Determination of the angle of attack on rotor blades, Wind Energy, 12, 91–98, doi:10.1002/we.277, 2009. 10

Sherry, M., Sheridan, J., and Jacono, D.: Characterisation of a horizontal axis wind turbine's tip and root vortices, Exp. Fluids, 54, 1417, doi:10.1007/s00348-012-1417-y, 2013. 3, 13

Sicot, C., Devinant, P., Loyer, S., and Hureau, J.: Rotational and turbulence effects on a wind turbine blade. Investigation of the stall mechanisms, J. Wind Eng. Ind. Aerod., 96, 1320–1331, doi:10.1016/j.jweia.2008.01.013, 2008. 3, 11

Snel, H., Houwink, R., van Bussel, G., and Bruining, A.: Sectional prediction of 3D effects for stalled flow on rotating blades and comparison with measurements, in: 1993 European Community Wind Energy Conference Proceedings, James & James 395–399, Travemünde, Germany, 1993. 3

Troldborg, N., Sørensen, J. N., and Mikkelsen, R.: Actuator Line Simulation of Wake of Wind Turbine Operating in Turbulent Inflow, in: The Science of Making Torque from Wind 2007, Copenhagen, Denmark, 2007. 4

van Kuik, G. A. M., Micallef, D., Herraez, I., van Zuijlen, A. H., and Ragni, D.: The role of conservative forces in rotor aerodynamics, J. Fluid Mech., 750, 284–315, doi:10.1017/jfm.2014.256, 2014. 4, 12

Vermeer, L., Sørensen, J., and Crespo, A.: Wind turbine wake aerodynamics, Prog. Aerosp. Sci., 39, 467–510, doi:10.1016/S0376-0421(03)00078-2, 2003. 3, 13

**WESD**

doi:10.5194/wes-2015-1

**Analysis of the Blade Root Flow**

I. Herráez et al.

**WESD**

doi:10.5194/wes-2015-1

**Analysis of the Blade Root Flow**

I. Herráez et al.

**Table 1.** Airfoil type distribution along the blade span.

| Radial position range [$r/R$] | Airfoil type |
| --- | --- |
| 0.14–0.16 | Cylinder |
| 0.16–0.21 | Transition |
| 0.21–1.0 | DU96-W-180 |

Discussion Paper | Discussion Paper | Discussion Paper | Discussion Paper |

WESD

doi:10.5194/wes-2015-1

**Analysis of the Blade Root Flow**

I. Herráez et al.

Interactive Discussion

**Table 2.** $C_\mathrm{l}$ and $C_\mathrm{d}$ for the simulated 2-D airfoil and 3-D blade at $r = 0.26\,R$, AoA $\approx 13°$.

|      | $C_\mathrm{l}$ | $C_\mathrm{d}$ |
|------|------|------|
| 2-D  | 0.97 | 0.07 |
| 3-D  | 1.06 | 0.07 |

Discussion Paper | Discussion Paper | Discussion Paper | Discussion Paper | Discussion Paper |

**WESD**

doi:10.5194/wes-2015-1

**Analysis of the Blade Root Flow**

I. Herráez et al.

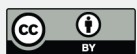

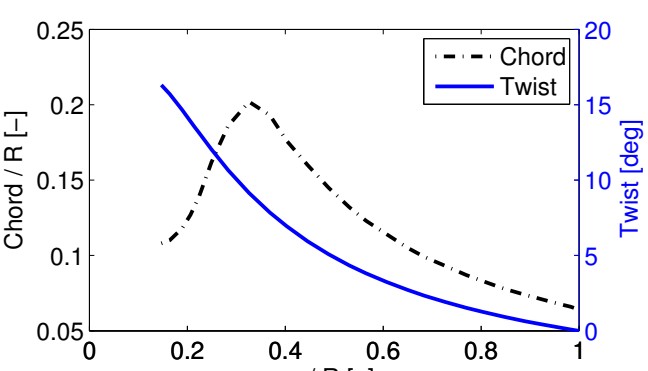

**Figure 1.** Chord and twist distribution along the blade span.

**WESD**

doi:10.5194/wes-2015-1

**Analysis of the Blade Root Flow**

I. Herráez et al.

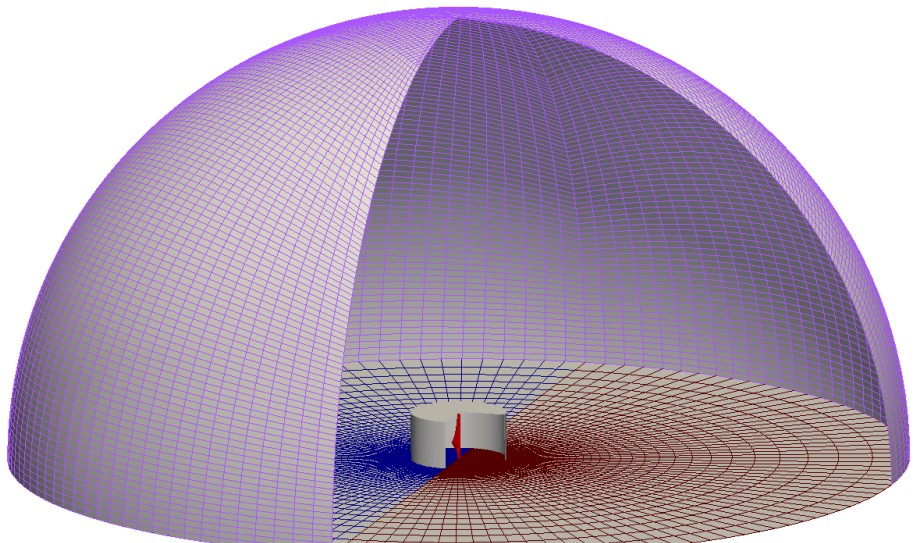

**Figure 2.** Schematic representation of the computational domain. The inner cylinder represents the arbitrary mesh interface.

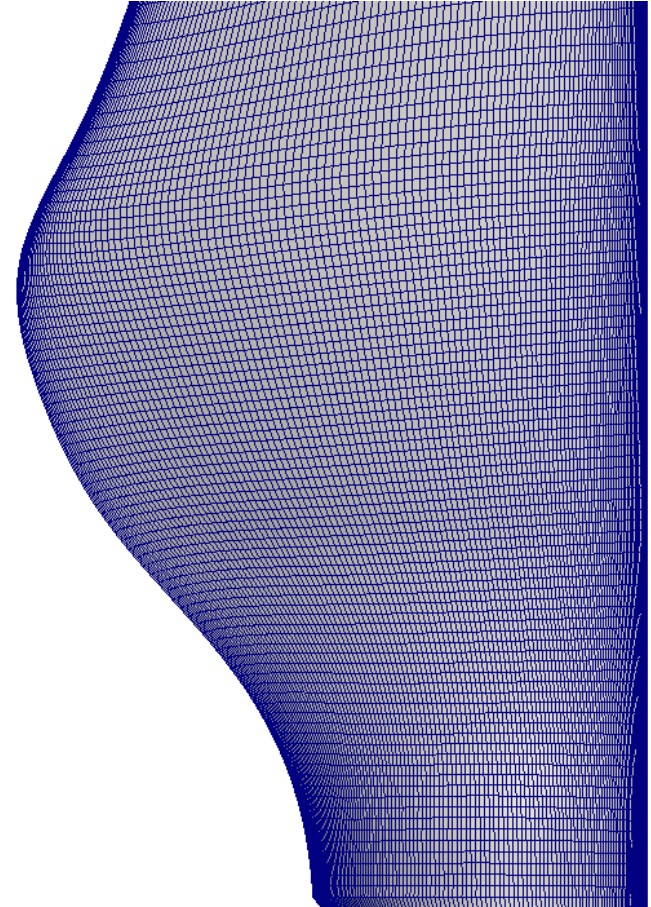

**Figure 3.** Detail of the surface mesh in the blade root region.

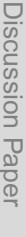

# WESD

doi:10.5194/wes-2015-1

**Analysis of the Blade Root Flow**

I. Herráez et al.

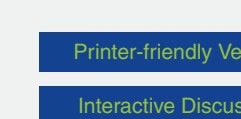

**WESD**

doi:10.5194/wes-2015-1

**Analysis of the Blade Root Flow**

I. Herráez et al.

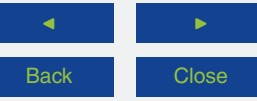

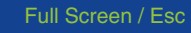

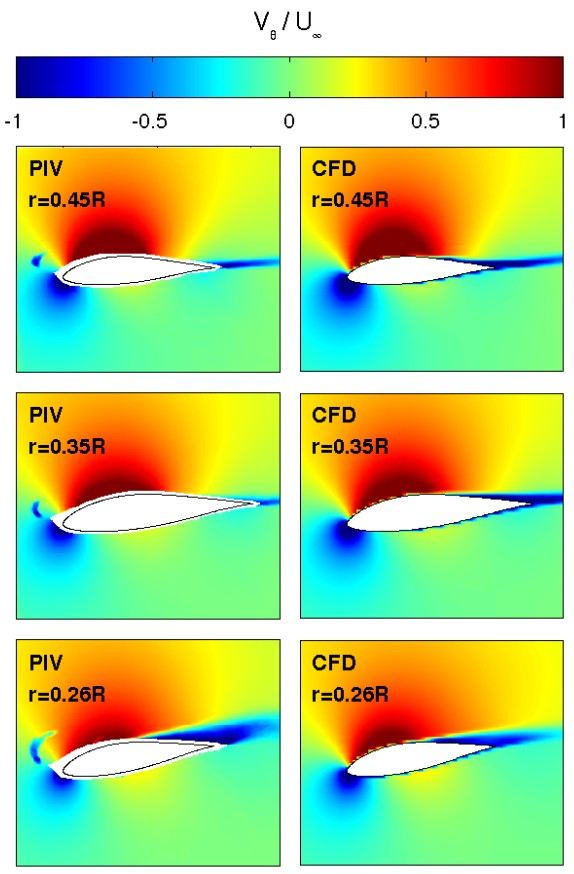

**Figure 4.** Experimental and numerical results of the azimuthal velocity component at different blade spanwise positions.

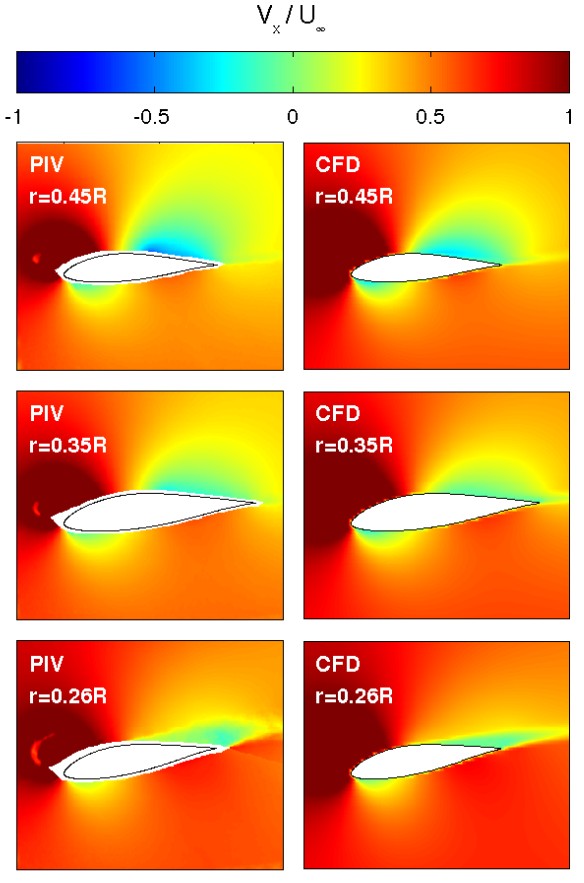

**Figure 5.** Experimental and numerical results of the axial velocity component at different blade spanwise positions.

**WESD**

doi:10.5194/wes-2015-1

**Analysis of the Blade Root Flow**

I. Herráez et al.

Discussion Paper | Discussion Paper | Discussion Paper | Discussion Paper | Discussion Paper |

**WESD**

doi:10.5194/wes-2015-1

**Analysis of the Blade Root Flow**

I. Herráez et al.

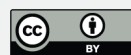

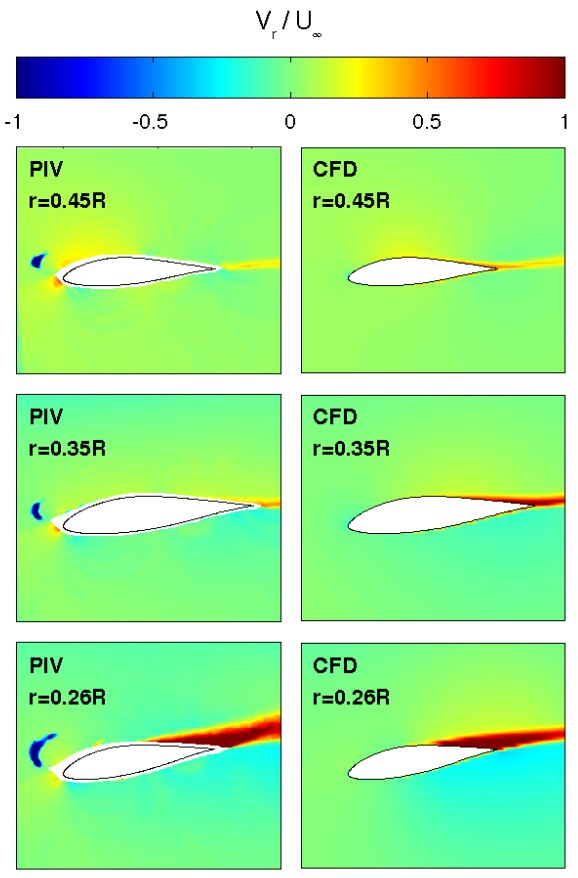

**Figure 6.** Experimental and numerical results of the radial velocity component at different blade spanwise positions.

**WESD**

doi:10.5194/wes-2015-1

**Analysis of the Blade Root Flow**

I. Herráez et al.

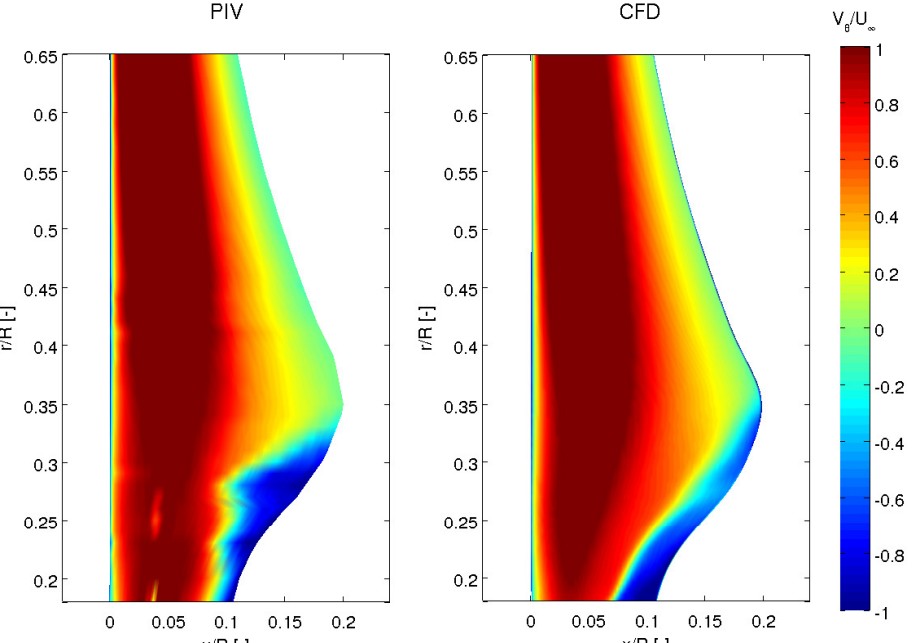

**Figure 7.** Experimental and numerical results of the azimuthal velocity component 10 mm off the blade suction side.

Discussion Paper | Discussion Paper | Discussion Paper | Discussion Paper | Discussion Paper |

**WESD**

doi:10.5194/wes-2015-1

**Analysis of the Blade Root Flow**

I. Herráez et al.

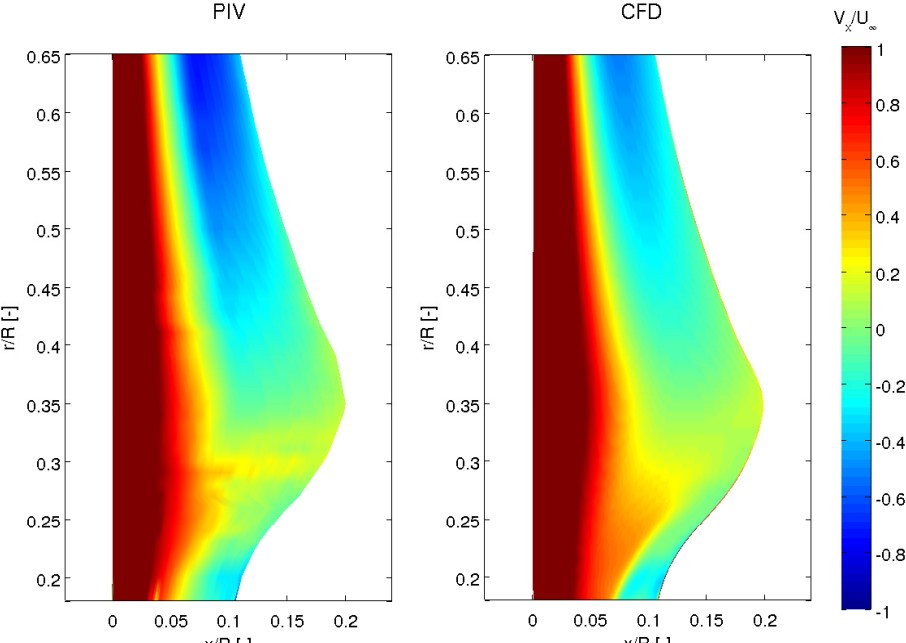

**Figure 8.** Experimental and numerical results of the axial velocity component 10 mm off the blade suction side.

Discussion Paper | Discussion Paper | Discussion Paper | Discussion Paper |

**WESD**

doi:10.5194/wes-2015-1

**Analysis of the Blade Root Flow**

I. Herráez et al.

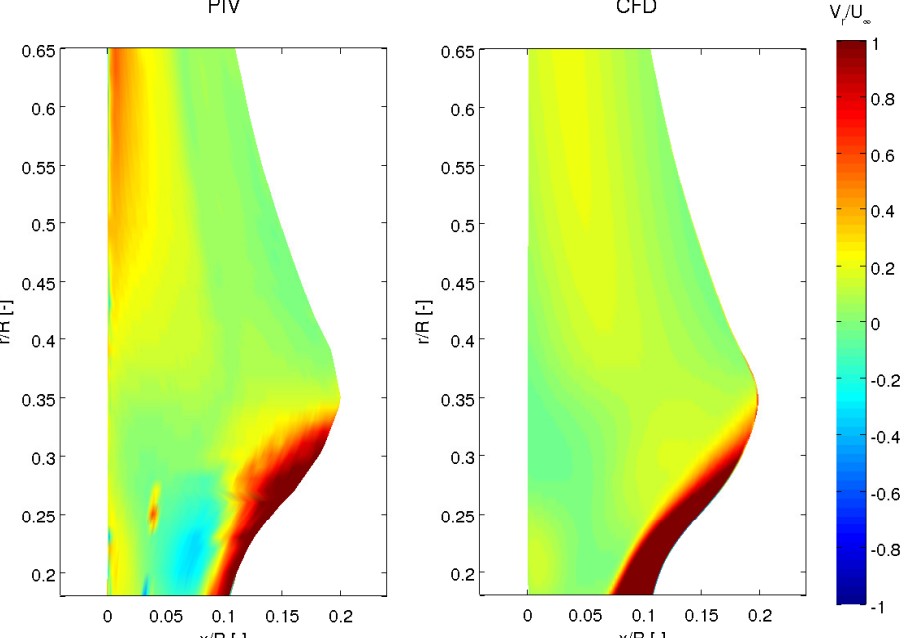

**Figure 9.** Experimental and numerical results of radial velocity component 10 mm off the blade suction side.

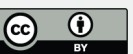

**WESD**

doi:10.5194/wes-2015-1

**Analysis of the Blade Root Flow**

I. Herráez et al.

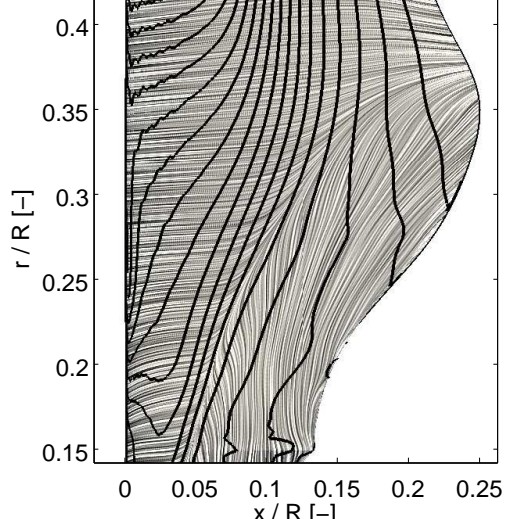

**Figure 10.** Isobars and limiting streamlines over the suction side of the blade root region.

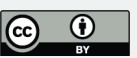

# WESD

doi:10.5194/wes-2015-1

**Analysis of the Blade Root Flow**

I. Herráez et al.

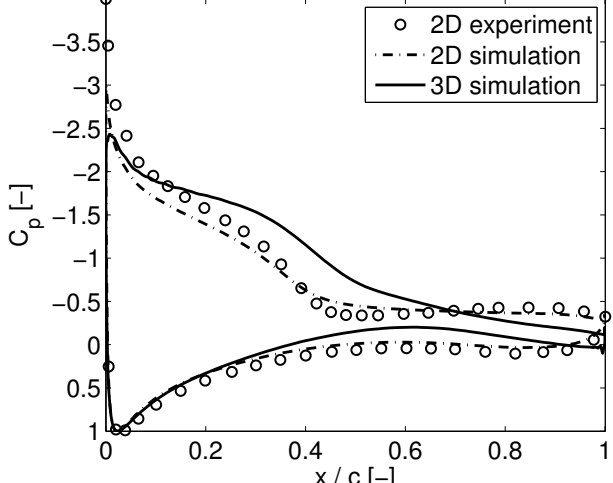

**Figure 11.** Cp distributions at AoA ≈ 13˚. The corresponding Reynolds numbers are $Re \approx 1 \times 10^6$ for the 2-D experimental results, $Re \approx 1 \times 10^5$ for the simulated 2-D airfoil and $Re \approx 1 \times 10^5$ for the 3-D blade ($r = 0.26\,R$).

**WESD**

doi:10.5194/wes-2015-1

**Analysis of the Blade Root Flow**

I. Herráez et al.

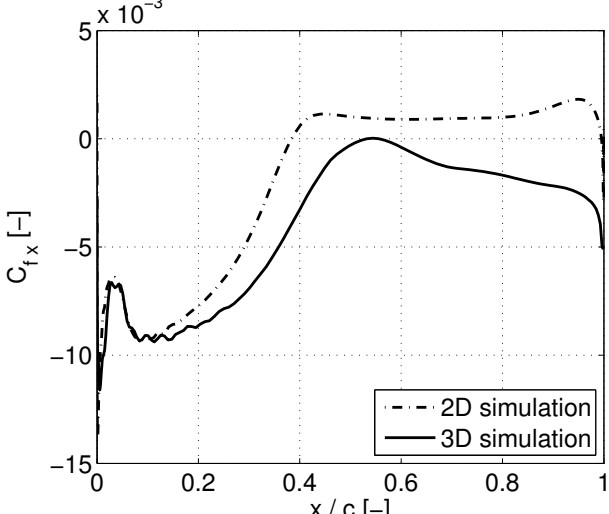

**Figure 12.** Wall shear stress in the chordwise direction over the suction side for the 2-D and 3-D ($r/R = 0.26$) cases.



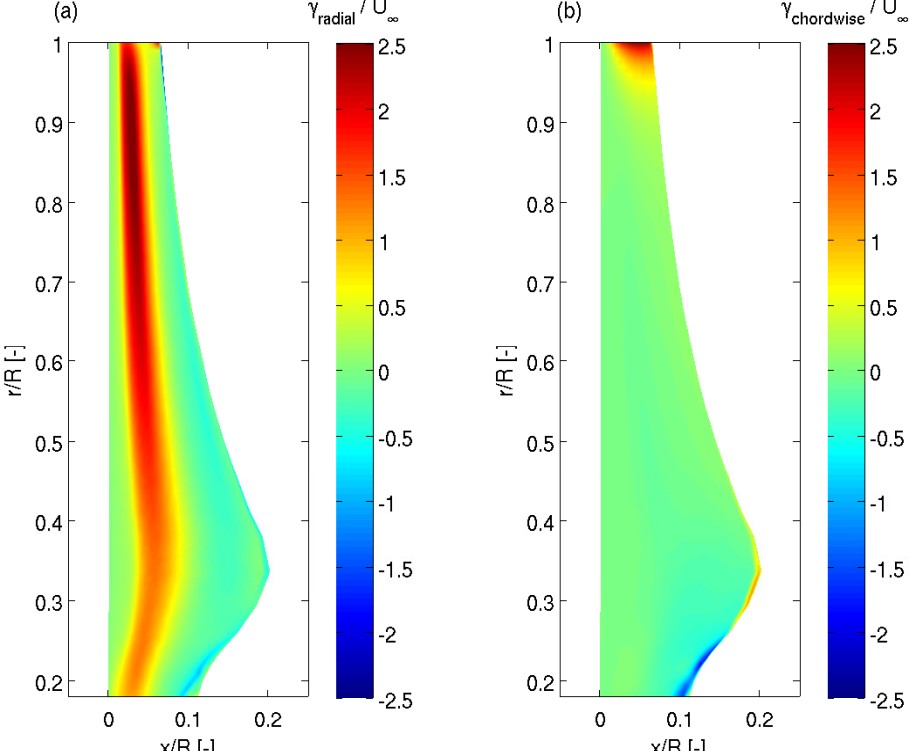

**Figure 13.** Radial and chordwise components of the bound vorticity.

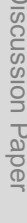

**WESD**

doi:10.5194/wes-2015-1

**Analysis of the Blade Root Flow**

I. Herráez et al.

## WESD

doi:10.5194/wes-2015-1

**Analysis of the Blade Root Flow**

I. Herráez et al.

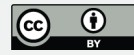

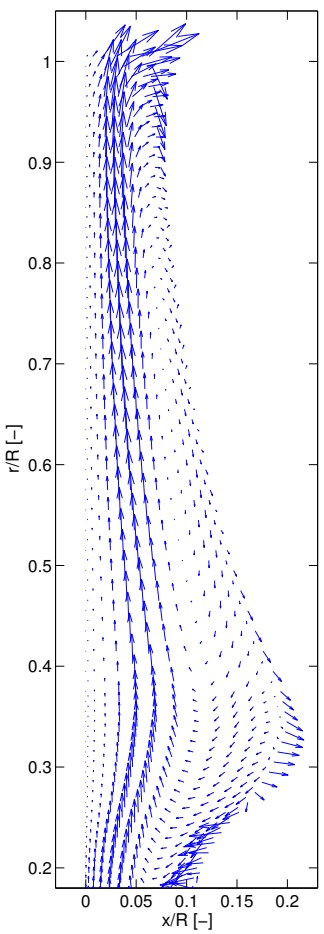

**Figure 14.** Bound vorticity vectors over the blade.