# Peer review of "Wind Energ. Sci. Discuss., doi:10.5194/wes-2015-1, 2016 Manuscript under review for journal Wind Energ. Sci. Published: 19 January 2016 © Author(s) 2016. CC-BY 3.0 License."

_Wind Energy Science, 2015_

## Referee Comment (RC1) · Anonymous Referee #1 · 15 Feb 2016

General comments: In this paper, the authors investigated the Himmelskamp effect and the origin of root vortex of a horizontal axis wind turbine by means of PIV and RANS. The RANS computed results show overall good agreement with the PIV measurements. The flow in the blade root region was analyzed in detail. The presented results are interesting and helpful for improving people's understanding of the dynamics of blade root flow and developing advanced correction models to account for the three-dimensional effects when employing two-dimensional airfoil data .

Specific comments: 1. Is the nacelle modeled in the simulation? 2. Figure 9 shows that the computed radial velocity is much smaller than that from measurements. The other possible reason could be that the RANS model is not very accurate in predicting such flows.
[Figure]

Technical corrections: 1. Is figure 10 from the RANS simulation or PIV measurements? 2. In figure 4, V_x denotes the axial velocity. On the other hand, x denotes chordwise direction in other figures, e.g. figure 10 and 11. This needs to be consistent. 3. Figures 4, 5 and 6: axes with titles, labels and ticks are needed.

————————————————————

---

## Referee Comment (RC2) · Anonymous Referee #2 · 22 Apr 2016

Review of paper WES doi:10.5194/wes-2015-1

Title: Detailed Analysis of the Blade Root Flow of a Horizontal Axis Wind Turbine Authors: I. Herraez et al.

The manuscript deals with an important area in wind turbine aerodynamics, i.e. rotational augmentation along the inboard blade sections of horizontal-axis wind turbine blades. The authors completed a very detailed PIV study of sectional- and blade-flow characteristics and compared against companying RANS analyses. Overall, good agreement is achieved between experimental and computational data. The manuscript merits publication in WES pending some comments given below.

General comments:

This is a good quality manuscript, and both results and interpretation appear to be correct. The reviewer enjoyed reading about the gamma (chordwise vorticity) distribution and its effect on the root vortex. Reading the abstract, though, there is not a single piece of information that is not already known to the wind energy community. In fact, a counter-rotating vortex has been documented for the NREL Phase VI rotor, resulting in a number of (also) computational studies that are not included in the manuscript.

- A sketch on a global coordinate system would be extremely helpful. The authors talk about radial, axial, and azimuthal but a clear sketch is missing. Then the equation about the chordwise vorticity distribution includes dx ?

- In reference to the previous comment, the authors may consider looking at a fairly recent paper by Dumitrache (AIAA J. Aircraft ?) that includes a very informative sketch (and coordinate definition) about rotational augmentation and the effects of both spanwise pressure gradients and centrifugal pumping.

- The authors need to address the scaling issue of rotational augmentation effects. In the end, this work, though very detailed for 1 operating case of 1 small-scale rotor, cannot be generalized, and this should be stated. Recent work by Lindenburg (ECN, Ph.D.) and Dowler and Schmitz (Wind Energy paper on BEM solution-based stall delay) identified a dimensionless parameter (ratio of centrifugal to Coriolis forces) that can help in quantifying the degree of rotational augmentation for this particular model rotor. It's very easy to add and would improve the paper.

- Also, the discussion on spanwise pressure gradient versus centrifugal pumping can be supported by, for example, the work of Du & Selig who did quite a nice analysis and provide the standard model still in use today in NREL codes. At least, the authors should mention this in the context of the discussion on pages 9-10.

- Check for typos, comma placement, etc.

Specific comments:
Page 4, Line 1: The rapid vortex diffusion could also be due to the low Reynolds numbers. Some discussion would be good.

Page 4, Line 24: Wording "... focus is put ..."

Page 5, Line 3: "The origin of the root vortex". (This is not Darwin's 'Origin of Species') Maybe something similar to "Root vortex formation in the presence of rotational augmentation"

Page 5, Line 14: Wording "... airfoil types"

Page 5, Lines 20-25: Here a clear sketch of the experimental setup is absolutely mandatory.

Page 6, Line 4-7: A figure showing baseline airfoil data (Re < 1x10^5) would be helpful. Again, it is unclear how results obtained are relevant to larger turbines. Look at the scaling parameter of centrifugal over Coriolis forces in recent works.

Page 6, Line 21: Add a full reference to Pointwise.

Page 7, 1st paragraph: For the RANS computations, wouldn't laminar-turbulent transition be of importance? At least it has to be addressed.

Page 7, Line 22: "excellent" is too strong of a statement without further justification and quantification.

Page 8, Line 12: Why? Laminar-turbulent transition?

Page 9, Line 3: Probably not the only source of uncertainty, others should be (at least) mentioned.

Page 9: Do the RANS computations include the blade hub ? I think this is potentially important.

---

## Author Comment (AC1) · 21 May 2016

**Reply to the comments of reviewer 1 on the manuscript wes-2015-1 "Detailed Analysis of the Blade Root Flow of a Horizontal Axis Wind Turbine"**

I. Herráez[1], B.Akay[2], G.J.W. van Bussel[2], J. Peinke[1,3] and B. Stoevesandt[3]

[1] Institute of Physics, University of Oldenburg, D-26111, Oldenburg, Germany
[2] Faculty of Aerospace Engineering, Delft University of Technology, 2629HS Delft, The Netherlands
[3] Fraunhofer IWES, Ammerländer Heerstr. 136, Oldenburg, Germany

We would like to thank the reviewer for his/her valuable comments and suggestions. In the following, we present our replies. The new version of the manuscript (including the corrections suggested by both referees) is appended to this document. Changes to the original manuscript have been highlighted in blue color.

*General comments:*

1. *Comment:* In this paper, the authors investigated the Himmelskamp effect and the origin of root vortex of a horizontal axis wind turbine by means of PIV and RANS. The RANS computed results show overall good agreement with the PIV measurements. The flow in the blade root region was analyzed in detail. The presented results are interesting and helpful for improving peoples understanding of the dynamics of blade root flow and developing advanced correction models to account for the three-dimensional effects when employing two-dimensional airfoil data
**Reply: Thank you very much for your overall positive assessment of our work.**

*Specific comments:*

1. *Comment: Is the nacelle modelled in the simulation?*
**Reply: The nacelle is not included in the numerical model. This information has been added to the new version of the manuscript.**

2. *Comment: Figure 9 shows that the computed radial velocity is much smaller than that from measurements. The other possible reason could be that the RANS model is not very accurate in predicting such flows.*
**Reply: Yes, actually both numerical and experimental uncertainties might play a role on this issue. This information has been included in the new version of the manuscript.**

*Technical corrections:*

1. *Comment: Is figure 10 from the RANS simulation or PIV measurements?*
**Reply: That figure shows only numerical results. This information has been added to new version of the manuscript.**

2. *Comment: In figure 4, $V_x$ denotes the axial velocity. On the other hand, x denotes chordwise direction in other figures, e.g. figure 10 and 11. This needs to be consistent.*
**Reply: This issue has been corrected**

3. *Comment: Figures 4, 5 and 6: axes with titles, labels and ticks are needed*
**Reply: Each plot of the mentioned figures includes already the corresponding radial position. Adding further dimensions (axes with titles, labels and ticks) does not contribute to a better understanding of the results and makes the plots more complicated. Therefore, we would prefer to leave these plots as they are for the sake of clarity.**

Manuscript prepared for Wind Energ. Sci.
with version 2015/11/06 7.99 Copernicus papers of the LaTeX class copernicus.cls.
Date: 21 May 2016

**Detailed Analysis of the Blade Root Flow of a Horizontal Axis Wind Turbine**

**Iván Herráez**[1]**, Busra Akay**[2]**, Gerard J. W. van Bussel**[2]**, Joachim Peinke**[1,3]**, and Bernhard Stoevesandt**[3]

[1]Institute of Physics, University of Oldenburg, D-26111, Oldenburg, Germany
[2]Faculty of Aerospace Engineering, Delft University of Technology, 2629HS Delft, The Netherlands
[3]Fraunhofer IWES, Ammerländer Heerstr. 136, Oldenburg, Germany

*Correspondence to:* Iván Herráez (ivan.herraez@forwind.de)

**Abstract.** The root flow of wind turbine blades is subjected to complex physical mechanisms that influence significantly the rotor aerodynamic performance. Spanwise flows, the Himmelskamp effect and the formation of the root vortex are examples of interrelated aerodynamic phenomena that take place in the blade root region. In this study we address those phenomena by means of Particle Image Velocimetry (PIV) measurements and Reynolds Averaged Navier-Stokes (RANS) simulations. The numerical results obtained in this study are in very good agreement with the experiments and unveil the details of the intricate root flow. The Himmelskamp effect is shown to delay the stall onset and to enhance the lift force coefficient $C_l$ even at moderate angles of attack. This improvement of the aerodynamic performance occurs in spite of the negative influence of the mentioned effect on the suction peak of the involved blade sections. The results also show that the vortex emanating from the spanwise position of maximum chord length rotates in the opposite direction of the root vortex, what affects the wake evolution. Furthermore, the aerodynamic losses in the root region are demonstrated to take place much more gradually than at the tip.

[revised manuscript text omitted]
 and is the one presented in this work. A sketch of the corresponding experimental set-up with the global coordinate system is displayed in Fig 2.

[Figure]

**Figure 2.** Experimental set-up with the chordwise PIV measurement configuration. The coordinate system used in this work is also displayed. The azimuthal direction $\theta$ is opposite to the direction of rotation $\omega$.

The measurements are phase-locked and phase-averaged with the azimuthal position of the rotor blade rotation. This allows to reconstruct the flow over each blade section after measuring the pressure and suction sides separately.

The rotor operated at rated conditions with a freestream wind speed $U_\infty = 6$ m/s and a rotational speed $\omega = 400$ rpm (tip speed ratio $\lambda = 7$). The turbulence intensity is $TI = 0.28\%$ and there is no yaw misalignment. The Reynolds number at the radial position of maximum chord reached $Re \approx 1.5 \times 10^5$.

Further details about the experimental set-up can be found in Akay et al. (2014).

**2.2 Numerical method and computational mesh**

The simulations presented in this work are based on the Reynolds-Averaged Navier-Stokes (RANS) method and they are performed with the open source code OpenFOAM (2015). The computational model solves the incompressible Navier-Stokes equations using a finite volume approach for the spatial discretization. The convective terms are discretized with a second order linear-upwind scheme. For the viscous terms a second-order central differences linear scheme is employed. The use of a non-inertial reference frame and the addition of the Coriolis and centrifugal forces to the momentum equations allows to account for the rotation of the system. The SIMPLE algorithm is employed for enforcing the pressure-velocity coupling. The turbulence in the boundary layer is modelled by means of the $k - \omega$ Shear-Stress Transport (SST) model proposed by Menter (1993). This model has been proved to be suitable for the simulation of wind turbine blades (Bechmann et al., 2011; Johansen and Sørensen, 2004; Le Pape and Lecanu, 2004; Sørensen et al., 2002). However, the implicit assumption of fully turbulent flow might be a source of uncertainty, since the existence of laminar to turbulent transition can not be completely ruled out.

The grid is generated with the software Pointwise (2015). The hub and nacelle geometries are disregarded in order to keep the mesh as simple as possible. This approach, which is based on the assumption that the hub and nacelle do not influence substantially the blade root flow, is usually followed when structured meshes are used for simulating wind turbine blades (Johansen et al., 2002; Sørensen et al., 2002; Le Pape and Lecanu, 2004; Schreck et al., 2007; Bechmann et al., 2011). The mesh exploits the symmetry of the rotor by modelling only one half of it and using periodic boundary conditions. The computational domain is represented in Fig. 4 and it consists of two independent block-structured grids connected by means of a so called arbitrary mesh interface. The outer grid is a semi-sphere with the radius 22R, where R is the blade radius. The inner grid, which contains the blade, is a cylinder with the radius 1.1R and the height 1.1R. The motivation for using two structured grids connected by an interface is to independently control the mesh resolution in the proximity of the blade and in the far field. The total number of cells is $9.8 \times 10^6$. The blade surface mesh (see Fig. 4) contains 130 cells along the chord, while 210 cells are used in the spanwise direction. In order to properly resolve the boundary layer, the height of the first cell in the normal direction to the blade surface is set to $5 \times 10^{-6}$ m, what ensures that Y+ is smaller than one along the whole blade.

The semi-spherical outer boundary employs a boundary condition that changes its behaviour depending on the direction of the flow: in regions where the flow goes in, it works

[Figure]

**Figure 3.** Schematic representation of the computational domain. The inner cylinder represents the arbitrary mesh interface.

[Figure]

**Figure 4.** Detail of the surface mesh in the blade root region

205 7, and they will be just neglected in the interpretation of the results.

Figure 5 shows the azimuthal velocity component in an inertial reference frame for the radial stations r=0.26R, r=0.35R (the position of maximum chord length) and 210 r=0.45R. The results are normalized with the free-stream wind speed $U_\infty$. The agreement between experimental and

[Figure]

**Figure 5.** Experimental and numerical results of the azimuthal velocity component at different blade spanwise positions.

like a Dirichlet boundary condition assuming a predefined value of the velocity field. In regions where the flow goes out, it enforces a zero gradient condition (Neumann condi-195 tion). The symmetry plane makes use of periodic boundary conditions. No-slip boundary conditions are applied to the blade surface.

**3    Results and discussion**

**3.1    Main characteristics of the flow field over the blades**

200 The detailed PIV and numerical results provide a good insight into the main flow characteristics over the blade root. It is important to note that the PIV data are partly affected by (sickle-shaped) reflection artefacts in front of the leading edge. Those artefacts are easily recognizable in Fig. 5, 6 and

numerical results is fairly good for all the studied radial positions. At r=0.35R and r=0.45R, the azimuthal velocities are positive over the whole suction side except in the trailing 215 edge region. However, at r=0.26R the suction side presents negative velocities from the mid-chord until the trailing edge. This does not necessarily imply flow separation and recirculation, though, since the relative velocity might still remain positive along the whole suction side. Indeed, at that 220 radial position the local circumferential velocity caused by the blade rotation is $1.82/U_\infty$, what implies that the flow remains attached up to $U_\theta = -1.82/U_\infty$. In Sect. 3.3 the lack of separation is demonstrated by means of the wall shear stress.

The axial flow component is shown in Fig. 6. The axial 225 velocity over the second half of the suction side becomes

[Figure]

[Figure]

[Figure]

**Figure 6.** Experimental and numerical results of the axial velocity component at different blade spanwise positions.

**Figure 7.** Experimental and numerical results of the radial velocity component at different blade spanwise positions.

smaller with increasing radial position. This is in fact just a geometrical effect that occurs as a consequence of the twist of the rotor blade. At r=0.45R the orientation of the second half of the suction side surface is slightly upstream, so the axial velocity becomes negative if the flow is attached. The effect is smaller for lower radial positions because of the larger twist angle, which neutralizes the mentioned geometrical effect. The numerical results are consistent with the experiments, although at r=0.45R the axial velocity over the suction side is slightly overpredicted.

Figure 7 displays the radial velocity component for the three considered spanwise positions. At r=0.26R, this velocity is very strong on the suction side. However, at r=0.35R and r=0.45R it becomes much weaker. Hence, the presence of spanwise flows seems to be limited to the innermost region of the blade. The agreement between experiments and simulations is again very good for all three stations.

The velocity field 10 mm off the suction side has been extracted from both the numerical results and the available PIV data (including 40 different radial positions between r=0.17R and r=0.65R) in order to study the flow in the proximity of the blade surface in more detail. Fig. 8 shows that the azimuthal component is always positive outboard of the position of maximum chord length (r=0.35R). Below that position, a significant region of the blade presents negative azimuthal velocities close to the trailing edge. At $r \approx 0.3R$, this effect is stronger in the PIV measurements than in the numerical results. Apart from that, the numerical results are in very good agreement with the experimental results.

The axial flow velocity component is displayed in Fig. 9. Outboard of the radial position of maximum chord length, the axial velocity becomes negative from the mid-chord towards the trailing edge. The same effect was already discussed in relation to Fig. 6. The fact that the numerical results underpredict a bit this effect, which is caused by the relative position of the suction side to the rotor plane, might indicate a possible small deviation in the pitch angle or some uncertainty in the PIV fields. For a detailed discussion of the experimental uncertainties the interested reader is referred to Akay et al. (2014). 
[revised manuscript text omitted]

**Table 2.** $C_l$ and $C_d$ for the simulated 2D airfoil and 3D blade at r=0.26R, AoA≈ 13°

|     | Cl   | Cd   |
| --- | ---- | ---- |
| 2D  | 0.97 | 0.07 |
| 3D  | 1.06 | 0.07 |

Estimating the validity of the above described results for larger wind turbines is not so straightforward, though. On one hand, the analysed turbine operates at the tip speed ratio $\lambda = \omega R/U_\infty = 7$, which is also realistic for full scale wind turbines working at nominal conditions. The NREL 5 MW wind turbine, for instance, also presents the same rated tip speed ratio (Jonkman et al., 2009). This is important because it contributes to maintain the same balance between the centrifugal and Coriolis forces, what is fundamental for the Himmelskamp effect. Lindenburg (2003), for instance, estimated that the change of aerodynamic lift and drag due to the Himmelskamp effect is proportional to the square of the local speed ratio $\omega r/U_{rel}$. Dowler and Schmitz (2015) also included a similar parameter, namely $2U_{rel}/\omega r$ (obtained directly from the ratio between the Coriolis and centrifugal forces) in their stall delay model. Interestingly, they also estimated the change of the lift force to be proportional to the square of the mentioned parameter.

On the other hand, the local blade solidity $c/r$, which has also also been identified as a fundamental parameter for the Himmelskamp effect (see e.g. Snel et al., 1993; Chaviaropoulos and Hansen, 2000; Lindenburg, 2003; Dowler and Schmitz, 2015), differs substantially between the TUDelft and the NREL 5M W turbines: at $r = 0.26R$ (i.e. the

[Figure]

**Figure 13.** Wall shear stress in the chordwise direction over the suction side for the 2D and 3D ($r = 0.26R$) cases.

[Figure]

**Figure 11.** Isobars and limiting streamlines over the suction side of the blade root region (obtained from the numerical results).

[Figure]

**Figure 12.** Cp distributions at AoA $\approx 13°$. The corresponding Reynolds numbers are $Re \approx 1 \times 10^6$ for the 2D experimental results, $Re \approx 1 \times 10^5$ for the simulated 2D airfoil and $Re \approx 1 \times 10^5$ for the 3D blade (r=0.26R).

tional to the square of the mentioned parameter. In any case, it can be inferred that the large discrepancy in the local blade solidity between both turbines would lead to a weaker Himmelskamp effect in the NREL 5MW turbine. This conclusion is also valid for other wind turbines of the same size, since they usually present a similar local blade solidity.

**3.4 The origin of the root vortex**

The bound vorticity $\gamma$ can be computed as the difference in the velocity outside the boundary layer of the pressure and suction sides. $\gamma$ can then be decomposed into a radial $\gamma_{radial}$ and a chordwise $\gamma_{chordwise}$ component. Figure 14 shows both components side by side.

$\gamma_{radial}$ is concentrated around the $1/4$ chord position, as might be expected. The radial circulation $\Gamma_{radial}$ can be computed from $\gamma_{radial}$ as its integral along the chord:

$$\Gamma_{radial} = \int_{le}^{te} \gamma_{radial} \cdot dx \tag{1}$$

where *le* and *te* are the leading and trailing edges, respectively. The use of the Kutta-Jukowski theorem allows then to compute the sectional lift:

$$L' = -\rho \cdot U_{rel} \cdot \Gamma_{radial} \tag{2}$$

where $\rho$ is the air density and $U_{rel}$ is the relative velocity in the plane perpendicular to the blade axis. Owing to the $\gamma_{radial}$ distribution from Fig. 14 and the strong link between $\gamma_{radial}$ and the lift, it can be concluded that the lift force is generated almost exclusively in the first half of the chord all along the blade. The decay of $\gamma_{radial}$, and hence the decay of the lift force, is much more sudden at the tip than at

radial position studied in Fig. 12 and 13) $c/R \approx 0.15$ for the TU-Delft and $c/R \approx 0.07$ for the NREL 5MW turbine. For some authors (e.g. Chaviaropoulos and Hansen, 2000; Lindenburg, 2003), the change in lift force is linearly proportional to the $c/r$ parameter, whereas for other authors (e.g. Snel et al., 1993; Dowler and Schmitz, 2015) it is propor-

[revised manuscript text omitted]

Le Pape, A. and Lecanu, J.: 3D Navier–Stokes computations of a stall–regulated wind turbine, Wind Energy, 7, 309–324, doi:10.1002/we.129, 2004.

Leishman, J. G.: Challenges in modelling the unsteady aerodynamics of wind turbines, Wind Energy, 5, 85–132, doi:10.1002/we.62, 2002.

Lindenburg, C.: Investigation into rotor blade aerodynamics, Tech. Rep. ECN-C03-025, ECN, Petten, Netherlands, 2003.

Massouh, F. and Dobrev, I.: Exploration of the vortex wake behind of wind turbine rotor, Journal of Physics: Conference Series, 75, 2007.

Medici, D. and Alfredsson, P. H.: Measurements on a wind turbine wake: 3D effects and bluff body vortex shedding, Wind Energy, 9, 219–236, doi:10.1002/we.156, http://dx.doi.org/10.1002/we.156, 2006.

Menter, F.: Zonal two equation k–$\omega$ turbulence models for aerodynamic flows, AIAA Journal, 93, 1993.

Micallef, D.: 3D flows near a HAWT rotor: A dissection of blade and wake contributions, Ph.D. thesis, Delft University of Technology, 2012.

Micallef, D., Akay, B., Ferreira, C. S., Sant, T., and van Bussel, G.: The origins of a wind turbine tip vortex, in: The Science of Making Torque from Wind 2012, Oldenburg, Germany, 2012.

Nilsson, K., Shen, W. Z., Sørensen, J. N., Breton, S.-P., and Ivanell, S.: Validation of the actuator line method using near wake measurements of the MEXICO rotor, Wind Energy, 18, 1683–1683, doi:10.1002/we.1864, http://dx.doi.org/10.1002/we.1864, 2015.

OpenFOAM: OpenFOAM: the open source CFD toolbox, www.openfoam.com, Accessed: October 2015, 2015.

Pointwise: Pointwise Inc, version 17.3R2, www.pointwise.com, Accessed: October 2015, 2015.

Raj, N.: An improved semi–empirical model for 3–D post–stall effects in horizontal axis wind turbines, Master's thesis, University of Illinois, Urbana–Champaign, 2000.

Ronsten, G.: Static pressure measurements on a rotating and a non-rotating 2.375 m wind turbine blade. Comparison with 2D calculations, Journal of Wind Engineering and Industrial Aerodynamics, 39, 105–118, doi:10.1016/0167-6105(92)90537-K, 1992.

Schepers, J.: Engineering models in wind energy aerodynamics, Ph.D. thesis, TU-Delft, 2012.

Schreck, S. and Robinson, M.: Rotational augmentation of horizontal axis wind turbine blade aerodynamic response, Wind Energy, 5, 133–150, doi:10.1002/we.68, 2002.

Schreck, S., Sant, T., and Micallef, D.: Rotational Augmentation Disparities in the MEXICO and UAE Phase VI Experiments, in: The Science of Making Torque from Wind 2010, Heraklion, Crete, Greece, 2010.

Schreck, S. J., Sørensen, N. N., and Robinson, M. C.: Aerodynamic structures and processes in rotationally augmented flow fields, Wind Energy, 10, 159–178, doi:10.1002/we.214, 2007.

Shen, W. Z., Hansen, M. O. L., and Sørensen, J. N.: Determination of the angle of attack on rotor blades, Wind Energy, 12, 91–98, doi:10.1002/we.277, 2009.

Sherry, M., Sheridan, J., and Jacono, D.: Characterisation of a horizontal axis wind turbine's tip and root vortices, Experiments in Fluids, 54, 1417, doi:10.1007/s00348-012-1417-y, http://dx.doi.org/10.1007/s00348-012-1417-y, 2013.

Sicot, C., Devinant, P., Loyer, S., and Hureau, J.: Rotational and turbulence effects on a wind turbine blade. Investigation of the stall mechanisms, Journal of Wind Engineering and Industrial Aerodynamics, 96, 1320–1331, doi:10.1016/j.jweia.2008.01.013, 2008.

Snel, H., Houwink, R., van Bussel, G., and Bruining, A.: Sectional prediction of 3D effects for stalled flow on rotating blades and comparison with measurements, in: 1993 European Community Wind Energy Conference Proceedings, pp. 395–399, Travemünde, Germany, 1993.

Sørensen, N. N., Michelsen, J. A., and Schreck, S.: Navier–Stokes predictions of the NREL phase VI rotor in the NASA Ames 80 ft × 120 ft wind tunnel, Wind Energy, 5, 151–169, doi:10.1002/we.64, 2002.

Troldborg, N., Sørensen, J. N., and Mikkelsen, R.: Actuator Line Simulation of Wake of Wind Turbine Operating in Turbulent Inflow, in: The Science of Making Torque from Wind 2007, Copenhagen, Denmark, 2007.

van Kuik, G. A. M., Micallef, D., Herraez, I., van Zuijlen, A. H., and Ragni, D.: The role of conservative forces in rotor aerodynamics, Journal of Fluid Mechanics, 750, 284–315, doi:10.1017/jfm.2014.256, 2014.

Vermeer, L., Sørensen, J., and Crespo, A.: Wind turbine wake aerodynamics, Progress in Aerospace Sciences, 39, 467 – 510, doi:http://dx.doi.org/10.1016/S0376-0421(03)00078-2, 2003.

---

## Author Comment (AC2) · 21 May 2016

**Reply to the comments of reviewer 2 on the manuscript wes-2015-1 "Detailed Analysis of the Blade Root Flow of a Horizontal Axis Wind Turbine"**

I. Herráez1, B.Akay2, G.J.W. van Bussel2, J. Peinke1,3 and B. Stoevesandt3

1 Institute of Physics, University of Oldenburg, D-26111, Oldenburg, Germany

2 Faculty of Aerospace Engineering, Delft University of Technology, 2629HS Delft, The Netherlands

3 Fraunhofer IWES, Ammerländer Heerstr. 136, Oldenburg, Germany

We would like to thank the reviewer for his/her valuable comments and suggestions. In the following, we present our replies. The new version of the manuscript (including the corrections suggested by both referees) is appended to this document. Changes to the original manuscript have been highlighted in blue color.

Comment: The manuscript deals with an important area in wind turbine aerodynamics, i.e. rotational augmentation along the inboard blade sections of horizontal-axis wind turbine blades. The authors completed a very detailed PIV study of sectional- and blade- flow characteristics and compared against companying RANS analyses. Overall, good agreement is achieved between experimental and computational data. The manuscript merits publication in WES pending some comments given below.

Reply: Thank you very much for your overall positive assessment of our work.

General comments:

- Comment: This is a good quality manuscript, and both results and interpretation appear to be correct. The reviewer enjoyed reading about the gamma (chordwise vorticity) distribution and its effect on the root vortex. Reading the abstract, though, there is not a single piece of information that is not already known to the wind energy community. In fact, a counter-rotating vortex has been documented for the NREL Phase VI rotor, resulting in a number of (also) computational studies that are not included in the manuscript.
   Reply: The abstract has been improved with additional information about the results. We could not find any article documenting a counter-rotating vortex in the root region of a wind turbine, but we would highly appreciate any specific hint in case the referee has more information about it.
- 2. Comment: A sketch on a global coordinate system would be extremely helpful. The authors talk about radial, axial, and azimuthal but a clear sketch is missing. Then the equation about the chordwise vorticity distribution includes dx ?

Reply: A sketch with the coordinate system together with the experimental set-up has been included in the new version of the manuscript (Figure 2).

- 3. Comment: In reference to the previous comment, the authors may consider looking at a fairly recent paper by Dumitrache (AIAA J. Aircraft ?) that includes a very informative sketch (and coordinate definition) about rotational augmentation and the effects of both spanwise pressure gradients and centrifugal pumping **Reply:** We could not find the article suggested by the reviewer but we would highly appreciate more detailed information about it.
- 4. Comment: The authors need to address the scaling issue of rotational augmentation effects. In the end, this work, though very detailed for 1 operating case of 1 small-scale rotor, cannot be generalized, and this should be stated. Recent work by Lindenburg (ECN, Ph.D.) and Dowler and Schmitz (Wind Energy paper on BEM solution-based stall delay) identified a dimensionless parameter (ratio of centrifugal to Coriolis forces) that can help in quantifying the degree of rotational augmentation for this particular model rotor. Its very easy to add and would improve the paper.

Reply: We have addressed this issue in the new version of the manuscript.

5. Comment: Also, the discussion on spanwise pressure gradient versus centrifugal pumping can be supported by, for example, the work of Du and Selig who did quite a nice analysis and provide the standard model still in use today in NREL codes. At least, the authors should mention this in the context of the discussion on pages 9-10.

**Reply: We have included the suggested reference of Du and Selig in the mentioned discussion.**

6. Comment: Check for typos, comma placement, etc. **Reply:** We have also improved the article in this regard.

**Specific comments:**

- Comment: Page 4, Line 1: The rapid vortex diffusion could also be due to the low Reynolds numbers. Some discussion would be good.
   Reply: Diffusion is indeed reduced at small Reynolds numbers. Therefore, we believe that the rapid vortex diffusion is related to the effect described in the manuscript.
- 2. Comment: Page 4, Line 24: Wording . . . focus is put . . . Reply: The text has been corrected according the to the reviewers' suggestion.
- Comment: Page 5, Line 3: 'The origin of the root vortex'. (This is not Darwin's 'Origin of Species') Maybe something similar to 'Root vortex formation in the presence of rotational augmentation'
   Reply: We have changed 'The origin of the root vortex' to 'The formation of the root vortex'.
- 4. Comment: Page 5, Line 14: Wording '. . . airfoil types' Reply: The text has been corrected according the to the reviewers' suggestion.
- 5. Comment: Page 5, Lines 20-25: Here a clear sketch of the experimental setup is absolutely mandatory Reply: The sketch of the set-up used in this work has been added to the new version of the manuscript (Fig. 2).
- 6. Comment: Page 6, Line 4-7: A figure showing baseline airfoil data ( $Re < 1 \times 10^5$ ) would be helpful. Again, it is unclear how results obtained are relevant to larger turbines. Look at the scaling parameter of centrifugal over Coriolis forces in recent works. Reply: No baseline airfoil data are available for the requested Reynolds number, so the suggested figure could not be added to the manuscript. The relevance of the current results for larger turbines is discussed in the new version of the article (also with regard to the ratio between Coriolis and centrifugal forces).
- 7. Comment: Page 6, Line 21: Add a full reference to Pointwise. Reply: The text has been corrected according the to the reviewers' suggestion.
- Comment: Page 7, 1st paragraph: For the RANS computations, wouldnt laminar-turbulent transition be of importance? At least it has to be addressed.
   Reply: This issue has been addressed in the new version of the manuscript.
- Comment: Page 7, Line 22: 'excellent' is too strong of a statement without further justification and quantification.
   Reply: We have changed the word 'excellent' by 'fairly good'.
- Comment: Page 8, Line 12: Why? Laminar-turbulent transition?
   Reply: The axial velocity component is not a good indicator of laminar to turbulent transition. As a consequence, it is not possible to infer out of the data that we have what is the source of the slight disagreement between PIV and CFD.
- 11. Comment: Page 9, Line 3: Probably not the only source of uncertainty, others should be (at least) mentioned. **Reply:** This issue has been addressed in the new version of the manuscript.
- 12. Comment: Page 9: Do the RANS computations include the blade hub ? I think this is potentially important. Reply: No, the numerical model does not include the hub. This information has been added to the manuscript.

Manuscript prepared for Wind Energ. Sci. with version 2015/11/06 7.99 Copernicus papers of the LATEX class copernicus.cls. Date: 21 May 2016

**Detailed Analysis of the Blade Root Flow of a Horizontal Axis Wind Turbine**

Iván Herráez1, Busra Akay2, Gerard J. W. van Bussel2, Joachim Peinke1,3, and Bernhard Stoevesandt3

1Institute of Physics, University of Oldenburg, D-26111, Oldenburg, Germany

2Faculty of Aerospace Engineering, Delft University of Technology, 2629HS Delft, The Netherlands 3Fraunhofer IWES, Ammerländer Heerstr. 136, Oldenburg, Germany

Correspondence to: Iván Herráez (ivan.herraez@forwind.de)

**Abstract.** The root flow of wind turbine blades is subjected to complex physical mechanisms that influence significantly the rotor aerodynamic performance. Spanwise flows, the Himmelskamp effect and the formation of the root vortex are examples of interrelated aerodynamic phenomena that take place in the blade root region. In this study we address those phenomena by means of Particle Image Velocimetry (PIV) measurements and Reynolds Averaged Navier-Stokes (RANS) simulations. The numerical results obtained in this study are in very good agreement with the experiments and unveil the details of the intricate root flow. The Himmelskamp effect is shown to delay the stall onset and to enhance the lift force coefficient  $C_l$  even at moderate angles of attack. This improvement of the aerodynamic performance occurs in spite of the negative influence of the mentioned effect on the suction peak of the involved blade sections. The results also show that the vortex emanating from the spanwise position of maximum chord length rotates in the opposite direction of the root vortex, what affects the wake evolution. Furthermore, the aerodynamic losses in the root region are demonstrated to take place much more gradually than at the tip.

30

[revised manuscript text omitted]
 and is the one pre- 150 sented in this work. A sketch of the corresponding experimental set-up with the global coordinate system is displayed in Fig 2.

**Figure 2.** Experimental set-up with the chordwise PIV measurement configuration. The coordinate system used in this work is also displayed. The azimuthal direction  $\theta$  is opposite to the direction of rotation  $\omega$ .

The measurements are phase-locked and phase-averaged with the azimuthal position of the rotor blade rotation. This allows to reconstruct the flow over each blade section after measuring the pressure and suction sides separately.

130

135

The rotor operated at rated conditions with a freestream wind speed  $U_{\infty} = 6$  m/s and a rotational speed  $\omega = 400$  rpm (tip speed ratio  $\lambda = 7$ ). The turbulence intensity is TI =

0.28% and there is no yaw misalignment. The Reynolds number at the radial position of maximum chord reached 190  $Re \approx 1.5 \times 10^5$ .

Further details about the experimental set-up can be found in Akay et al. (2014).

**2.2 Numerical method and computational mesh**

140

180

The simulations presented in this work are based on the Reynolds-Averaged Navier-Stokes (RANS) method and they are performed with the open source code OpenFOAM (2015). The computational model solves the incompressible Navier-Stokes equations using a finite volume approach for the spatial discretization. The convective terms are discretized with a second order linear-upwind scheme. For the viscous terms a second-order central differences linear scheme is employed. The use of a non-inertial reference frame and the addition of the Coriolis and centrifugal forces to the momentum equations allows to account for the rotation of the system. The SIMPLE algorithm is employed for enforcing the pressure-velocity coupling. The turbulence in the boundary layer is modelled by means of the  $k-\omega$  Shear-Stress Transport (SST) model proposed by Menter (1993). This model has been proved to be suitable for the simulation of wind turbine blades (Bechmann et al., 2011; Johansen and Sørensen, 2004; Le Pape and Lecanu, 2004; Sørensen et al., 2002). However, the implicit assumption of fully turbulent flow might be a source of uncertainty, since the existence of laminar to turbulent transition can not be completely ruled out.

The grid is generated with the software Pointwise (2015). The hub and nacelle geometries are disregarded in order to keep the mesh as simple as possible. This approach, which is based on the assumption that the hub and nacelle do not influence substantially the blade root flow, is usually followed when structured meshes are used for simulating wind turbine blades (Johansen et al., 2002; Sørensen et al., 2002; Le Pape and Lecanu, 2004; Schreck et al., 2007; Bechmann et al., 2011). The mesh exploits the symmetry of the rotor by modelling only one half of it and using periodic boundary conditions. The computational domain is represented in Fig. 4 and it consists of two independent block-structured grids connected by means of a so called arbitrary mesh interface. The outer grid is a semi-sphere with the radius 22R, where R is the blade radius. The inner grid, which contains the blade, is a cylinder with the radius 1.1R and the height 1.1R. The motivation for using two structured grids connected by an interface is to independently control the mesh resolution in the proximity of the blade and in the far field. The total number of cells is  $9.8 \times 10^6$ . The blade surface mesh (see Fig. 4) contains 130 cells along the chord, while 210 cells are used in the spanwise direction. In order to properly resolve the boundary layer, the height of the first cell in the normal direction to the blade surface is set to  $5 \times 10^{-6}$  m, what ensures that Y+ is smaller than one along the whole blade.

The semi-spherical outer boundary employs a boundary condition that changes its behaviour depending on the direction of the flow: in regions where the flow goes in, it works

**Figure 3.** Schematic representation of the computational domain. The inner cylinder represents the arbitrary mesh interface.